# Impact of Regional Block Failure in Ambulatory Hand Surgery on Patient Management: A Cohort Study

**DOI:** 10.3390/jcm9082453

**Published:** 2020-07-31

**Authors:** Lucile Picard, Pierre Belnou, Claire Debes, Nathanael Lapidus, Eileen Sung Tsai, Julien Gaillard, Alain Sautet, Francis Bonnet, Thomas Lescot, Franck Verdonk

**Affiliations:** 1Department of Anesthesiology and Intensive Care, Hôpital Saint-Antoine, Assistance Publique-Hôpitaux de Paris, 75012 Paris, France; lucile.picard@aphp.fr (L.P.); claire.debes@aphp.fr (C.D.); francis.bonnet@aphp.fr (F.B.); thomas.lescot@aphp.fr (T.L.); 2Unité de Santé Publique, Hôpital Saint-Antoine, Assistance Publique-Hôpitaux de Paris, 75012 Paris, France; pierre.belnou@aphp.fr (P.B.); nathanael.lapidus@upmc.fr (N.L.); 3Department of Anesthesiology, Perioperative and Pain Medicine, Stanford University School of Medicine, Stanford, CA 94305, USA; estsai@stanford.edu; 4Orthopedic Surgery Department, Hôpital Saint-Antoine, Assistance Publique-Hôpitaux de Paris, 75012 Paris, France; julien.gaillard@aphp.fr (J.G.); alain.sautet@aphp.fr (A.S.); 5Orthopedic Surgery Department, American Hospital of Paris, 92200 Neuilly-sur-Seine, France; 6School of Medicine, Sorbonne University, 75012 Paris, France

**Keywords:** regional anesthesia, ambulatory surgery, block failure, patient management, risk factors

## Abstract

Regional anesthesia (RA) is an anesthetic technique essential for the performance of ambulatory surgery. Failure rates range from 6% to 20%, and the consequences of these failures have been poorly investigated. We determined the incidence and the impact of regional block failure on patient management in the ambulatory setting. This retrospective cohort study includes all adult patients who were admitted to a French University Hospital (Hôpital Saint-Antoine, AP-HP) between 1 January 2016 and 31 December 2017 for unplanned ambulatory distal upper limb surgery. Univariate and stepwise multivariate analyses were performed to determine factors associated with block failure. Among the 562 patients included, 48 (8.5%) had a block failure. RA failure was associated with a longer surgery duration (*p* = 0.02), more frequent intraoperative analgesics administration (*p* < 0.01), increased incidence of unplanned hospitalizations (*p* < 0.001), and a 39% prolongation of Post-Anesthesia Care Unit (PACU) length of stay (*p* < 0.0001). In the multivariate analysis, the risk factors associated with block failure were female sex (*p* = 0.04), an American Society of Anesthesiologists (ASA) score > 2 (*p* = 0.03), history of substance abuse (*p* = 0.01), and performance of the surgery outside of the specific ambulatory surgical unit (*p* = 0.01). Here, we have documented a significant incidence of block failure in ambulatory hand surgery, with impairment in the organization of care. Identifying patients at risk of failure could help improve their management, especially by focusing on providing care in a dedicated ambulatory circuit.

## 1. Introduction

The objectives of ambulatory anesthesia are to guarantee both a rapid discharge home of the patient and a comfortable recovery. Regional anesthesia (RA) fits the commitments of ambulatory surgery by allowing: (a) a good postoperative analgesia; (b) a rapid postoperative recovery; and (c) a decrease in the incidence of postoperative nausea and vomiting (PONV) [1,2,3]. Increasing ambulatory surgery is a real public health objective that is supported in France by the latest circular (published in September 2015) from the General Direction of the Care Offer (DGOS). It recommends that, by 2020, 66% of surgical procedures should be performed on an ambulatory basis [4].

Several studies document the benefits of RA when compared to general anesthesia [2,3]. The meta-analysis of Richman et al., which included 19 randomized studies and 603 patients, found that RA provided significantly better postoperative analgesia (*p* < 0.001) and lower postoperative opioid consumption (*p* < 0.001). Furthermore, fewer side effects have been reported with RA (*p* < 0.001) when compared with systemic opioid administration after GA [2]. However, 6% to 20% of patients operated under RA failed to achieve a successful anesthesia [5,6]. The French Society of Anesthesiology and Intensive Care Medicine (SFAR) has defined RA failure as the need for an additional block, sedation, or general anesthesia (GA) [7]. The failure rate depends on the technique used, the expertise of practitioners, and certain patient characteristics that remain controversial [5,8,9]. For example, a BMI > 25 and an American Society of Anesthesiologists (ASA) score of 4 are significantly associated with a higher RA failure rate [5]. However, these studies are based on data obtained almost 20 years ago, most of them using nerve stimulation guidance. Since the rise of ultrasound guidance, practices have changed and many studies have shown benefits, particularly in terms of speed of execution and time to onset of sensory and motor block, compared to the nerve stimulator technique, whatever the block considered [10,11].

There are limited data concerning RA failure and its consequences in the setting of ambulatory surgery. Consequently, we sought to analyze the incidence of RA failure in a uniformized ambulatory surgery of the upper limb [7] and to determine its risk factors and its consequences on the patient’s management.

## 2. Experimental Section

### 2.1. Study Design and Patients

We performed a retrospective study in the ambulatory surgical unit of a French University Hospital (Hôpital Saint-Antoine, Assistance Publique-Hôpitaux de Paris (AP-HP), Paris). All patients older than 18 years admitted to the ambulatory center of Hôpital Saint-Antoine for emergency distal orthopedic surgery (whitlow, smashed finger, acute fingernail injuries, or distal wound) between 1 January 2016 and 31 December 2017 were included. Patients who presented for another associated surgical procedure, had another technique of anesthesia, or had missing data in the main criteria were excluded.

### 2.2. Patient Management

In our institution, the center for ambulatory surgery is a unit of 3 operating rooms separated from the main operating theatre. It includes one preoperative room (with 6 armchairs) where RA is performed under monitoring, three operating rooms, and one Post-Anesthesia Care Unit (PACU), which also serves as a room for retrieving criteria for aptitude before the patient returns home.

All patients who have had hand trauma surgery are convened to be operated on the next morning after the trauma at 7 a.m. If the operating program is full in the unit, the patients are transferred to the conventional surgical center throughout the day for the procedure and are allowed to return home on the same day. All practitioners involved in patient care work equally in these two structures.

Anesthesia consisted of an ultrasound-guided axillary coupled with subcutaneous injection to anesthetize the medial cutaneous nerve in the case of tourniquet use (AxB), truncular (median, ulnar, or radial), digital intrathecal, or associated AxB and truncular block, depending on the experience of the anesthesiologist. The local anesthetic used was lidocaine or ropivacaine, depending on the need for prolonged postoperative analgesia. The blocks were tested before transfer to the operating room, following international guidelines, by testing different sensitive territories using light touch assessment [12].

### 2.3. Study Outcomes

The primary outcome was the incidence of block failure, defined as the need to supplement anesthesia during the intraoperative period. This anesthesia could either be sedation or general anesthesia or a new anesthetic block, as decided by each anesthesiologist in case it was impossible to achieve the surgery without it. [7]. The secondary outcomes included duration of the anesthesia and surgical procedures, the PACU length-of-stay, and the incidence of unscheduled hospitalization. We also determined the risk factors associated with block failure using patient demographic data extracted from their medical records: sex; age; ASA score; significant medical history and substance abuse defined as chronic drug abuse, including opioids; anesthetic and surgical characteristics such as the need for anxiolytic premedication; waiting time before RA (defined as the time from the patient’s arrival in the surgical unit to the realization of the anesthesia); type of regional block; and the use of intraoperative analgesics.

### 2.4. Regulatory and Ethical Aspects

In accordance with French law on biomedical research, this observational retrospective study obtained the approval of an Institutional Review Board (“Comité d’Éthique de la Recherche en Anesthésie-Réanimation” (CERAR, President Prof. JE Bazin, 27/05/19)) under the reference Institutional Review Board (IRB) 00010245-2019-086 [13]. Patients were all informed via the websites of the institution (AP-HP and Hôpital Saint-Antoine) of the possible use of their data in research aimed towards improving the quality of care, as well as their rights and terms of objection. This information was also included for each patient in the hospital’s welcome booklet, which was given during administrative registration and presented at the end of the hospitalization reports.

In order to guarantee the security of personal data, the investigators retrospectively collected and integrated the information anonymously into a secure database in accordance with the French Commission nationale de l’informatique et des libertés (CNIL) reference methodology (MR) - 004 and registered it in the AP-HP processing register under number 20190813142124.

### 2.5. Statistical Methodology

A preliminary sample size calculation assessed that about 50 events would be necessary to identify the main risk factors with sufficient power. After two years of recruitment, our sample size (48 patients with block failure and 514 without) allowed us to identify an Odds Ratio (OR) of about 2.5 for a 20% to 50% proportion of exposed controls with 80% power, which was considered enough to end recruitment. The characteristics of the study population are described on average (standard deviation) for quantitative variables and in number (percentage) for categorical variables. Inclusion characteristics were compared between subjects who underwent block failure and those who did not, using Mann–Whitney–Wilcoxon tests for quantitative variables with a non-normal distribution and Fisher tests for categorical variables. Variables of clinical relevance or associated with RA failure at the 0.20 significance threshold were selected in a multivariate logistic regression model selection procedure. The association forces are expressed in Odds Ratios (ORs) with their 95% confidence interval. The selected model proposed only significant associations with RA failure (*p* < 0.05) and minimized the Akaike Information Criterion (AIC). The maximum number of estimated parameters was limited to 5 in view of the number of events observed.

The average duration of management was compared between these two groups in survival analysis using the Kaplan–Meir method. For example, the available numbers and the observed proportion of patients with RA failure showed an OR of about 2.5 for a proportion of controls exposed from 20% to 50% with a power of 80%. All tests were performed at a significance level of 0.05. The analyses were carried out using the software R version 3.6 (R Foundation for Statistical Computing, Vienna, Austria).

## 3. Results

### 3.1. Characteristics of the Study Population

During the study period between 1 January 2016 and 31 December 2017, 647 patients had hand surgery. Sixteen patients were less than 18 years old and not included. Sixty-nine (10%) other patients were excluded. More specifically, 8 (1%) patients received first-line general anesthesia, 12 (2%) patients had another associated surgery, and 49 (7%) patients had missing data. A total of 562 patients were eventually analyzed (Figure 1).

The characteristics of the study population are summarized in Table 1. Among the 562 patients, 48 patients (8.5%) had block failure. Among the 48 patients in the failure group, 29 patients (60%) required additional sedation, 17 patients (35%) had general anesthesia, and 2 patients (5%) had a supplemental block because of the inability to perform surgery with only the first block.

### 3.2. Patient Management after Block Failure

Block failure was associated with a longer duration of surgery (35 +/− 28 min vs. 26 +/− 18 min in patients with and without block failure, respectively (*p* = 0.02)). These same patients also benefited from more frequent administration of intraoperative analgesics (7 patients (14.6%) in the failure group vs. 18 patients (3.5%) in the success group (*p* < 0.01)).

Only one patient in the failure group complained of nausea and vomiting after surgery, compared to three patients in the other group (*p* = 0.30). In the failure group, 11 patients (23%) required the administration of analgesics in the Post-Anesthesia Care Unit (PACU) vs. 144 patients (28%) in the success group (*p* = 0.85).

The length of stay in the PACU was 39% longer in the failure group in comparison to the success group (45 min vs. 32 min, respectively, *p* < 0.0001). In total, the median discharge time was evaluated at 120 min in the failure group vs. 100 min in the success group (Hazard Ratio (HR) = 0.61, 95% Confidence Interval (IC 95%) (0.45–0.82) *p* = 0.001) (Figure 2).

In the failure group, there was also an increase in the incidence of unscheduled hospitalizations with an incidence of 16% of hospitalization versus 4% in the success group (*p* < 0.01) (Table 2). Among the 30 hospitalized patients, 24 (80%) were operated on at night, between 6 p.m. and 12 a.m., 1 patient (3.3%) in the morning between 8 a.m. and 12 p.m., and finally 5 patients were operated on in the afternoon between 12 p.m. and 6 p.m. (Table 3).

### 3.3. Risk Factors Associated with Block Failure

After univariate analysis, the risk factors significantly associated with block failure were female sex (*p* = 0.04), an ASA score > 2 (*p* = 0.03), a history of substance abuse (*p* < 0.001), and the performance of surgery outside the ambulatory care unit (*p* = 0.01).

The multivariate model analysis identified the same independent risk factors. Female sex (Odds Ratio (OR) 2.35, CI 1.24-4.45, *p* = 0.01), the performance of surgery in a conventional surgical unit (OR 2.33, CI 1.25-4.34, *p* = 0.01), a history of substance abuse (OR 7.27 IC 1.18-44.88, *p* = 0.03) or a higher ASA score (OR 2.02 IC 1.01-3.94, *p* = 0.04) were risk factors for RA failure (Table 4).

## 4. Discussion

The incidence of block failure is relatively high in ambulatory hand surgery and associated with a longer length of hospital stay. The main risk factors found were female sex, a history of substance abuse, a higher ASA score, and the performance of surgery and anesthesia outside the ambulatory circuit. This incidence of block failure is in line with the data in the literature, which vary between 6% and 20% depending on the type of surgery considered [5]. To our knowledge, our study is the first to look at block failure since the rise of ambulatory care and ultrasonography and at its consequences in terms of patient flow and unplanned hospitalizations.

Failure was defined as the need for sedation, the addition of a block supplement, or a conversion to general anesthesia, according to recommendations of our national scientific society [7]. This definition has some limitations related to the inter-operator variability of the assessment of intraoperative block failure and could induce detection biases. However, given the fact that the anesthesiologist practice is homogenous in one center, this detection bias exists but should be considered marginal in this study. Nevertheless, it must be taken into account when considering the variability of the rate of failure among different studies.

The incidence of failure may depend on the type of block performed and the technique of performance. In the current study, axillary block was performed in half of the patients because of the sensitive territories involved in surgeries and to allow a good tolerance of the tourniquet. This block is not especially difficult to perform under ultrasound guidance, but the extent of the anesthetic solution may be limited and one or more nerves that are involved in the surgical area may be excluded. In other studies, the incidence of failure after axillary block has been reported to be as high as 17% despite ultrasound guidance [14]. In the case of hand surgery, failure may be due to the mechanism cited above but also to the decreasing tolerance of the tourniquet as the duration of the surgical procedure increases. This might be an interpretation of the association between duration of surgery and the risk of failure.

In the same way, the time of surgery seems to be a prognostic factor of block efficacy. In our study, 45 patients (8%) were operated on outside of the regular hours; they account for more than 80% of hospitalized patients (failure of ambulatory management). The block failure was 12% when surgery was performed at night, even if surgery was performed on an outpatient basis. Beyond the statistical question, this means the number of patients that need to be treated (i.e., switched from a conventional to an ambulatory surgical unit) to avoid one additional failure would be 2.9 (95% CI: 1.4, 7.5). We suggest that operating on ambulatory patients outside of regular hours may increase the postoperative morbidity, as already described for other procedures [15,16].

Obesity, which is also associated with block failure, is a common explanation given for technical difficulties, even with the support of ultrasound guidance [17].

Substance abuse has been identified as a factor in block failure. One hypothesis is that anxiety, a common feature in these patients, may change pain perception [18], a source of dissatisfaction. Several distraction methods have been shown to be useful in reducing anxiety and pain and improving patient satisfaction, and could be worth using in these patients [19,20,21].

Ultimately, the expertise of the operator may explain a difference in efficacy [22,23], although the learning curve of axillary block to achieve technical proficiency is rapid. In agreement with this, we failed to document that such a relationship exists between the success of the block and the expertise of the anesthetist.

Lastly, the implementation of RA and surgery in a dedicated ambulatory surgery area has been identified in our study as a protective factor against block failure, perfectly in agreement with current outpatient surgery recommendations [24]. This result supports the fact that, in addition to the technical aspects, the environmental conditions matter in the success of a procedure [25].

A meta-analysis by Liu et al. evaluated the efficacy of regional anesthesia compared to general anesthesia in the setting of ambulatory surgery [3], highlighting the fact that regional blocks allowed a faster recovery with a faster hospital discharge, a decrease in side effects such as postoperative nausea and vomiting, and sedation with better analgesic management. To our knowledge, no recent study has pointed out the consequences of RA failure on patient management during ambulatory surgery. In our study, block failure conveys a risk of longer hospital stay and is associated with a longer surgical procedure duration. Additional time devoted to anesthetic management could explain this fact. Our study also documented a 39% longer duration of stay in the PACU in the case of block failure, which can be easily explained by an increased sedation demand or the shift to general anesthesia.

This study is a retrospective, single center, observational study with possible bias. We included, however, a significant number of patients; the surgical procedures were standardized and performed routinely, making the cohort of patients quite homogeneous; and the collection of perioperative data was exhaustive.

As a limitation, patients’ anesthetic management, including the use of loco-regional anesthesia, may vary from one country to another, limiting the extrapolation of our data. For example, in some countries, supraclavicular and infraclavicular blocks are currently contraindicated in ambulatory settings. On another note, precise investigation into sensitive territories responsible for Regional Anesthesia RA failure would be useful to add some recommendations to our study, but as all the blocks were performed under the supervision of experienced anesthesiologists, we instead analyzed the implicated perioperative factors.

## 5. Conclusions

The rate of block failure is not negligible during ambulatory hand surgery and may be higher than expected by practitioners. The risk factors of block failure depend on patient and environmental conditions, and the main consequence is a prolongation of hospital stay. It could be of worth to evaluate the risk factors of block failure before anesthesia in order to either promote an alternative technique or manage the patients more carefully.

## Figures and Tables

**Figure 1 jcm-09-02453-f001:**
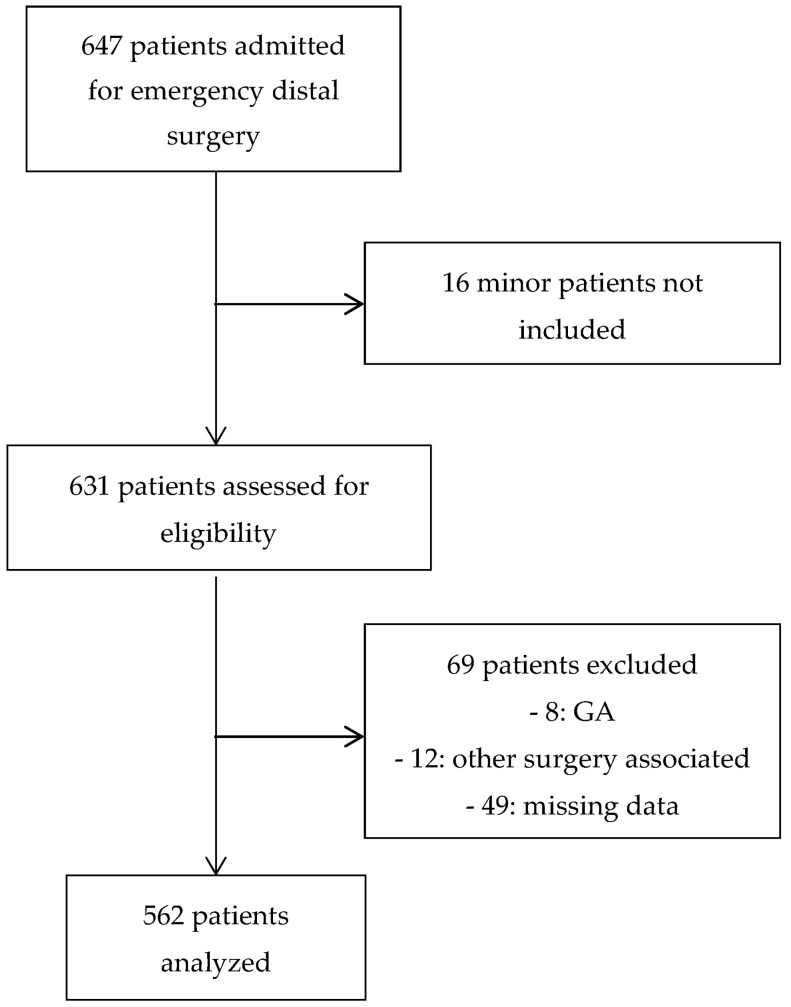
Flow chart of the study population.

**Figure 2 jcm-09-02453-f002:**
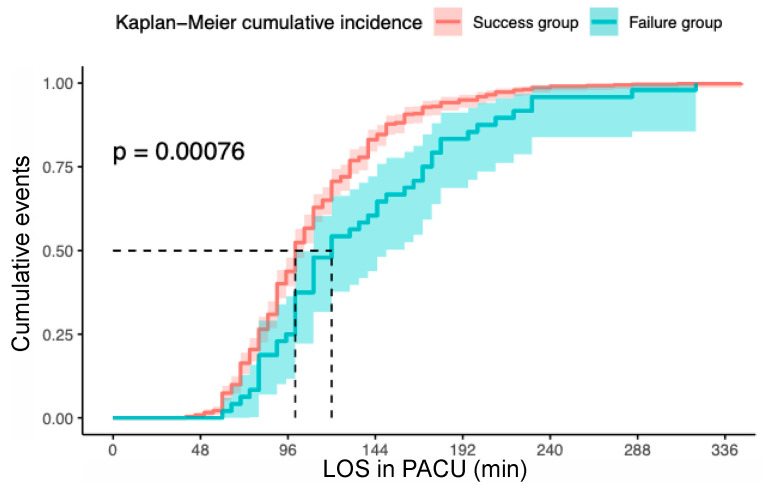
Kaplan–Meir estimation of the cumulative incidence of return to the street by overall care time. Abbreviations: LOS, length of stay.

**Table 1 jcm-09-02453-t001:** Demographic data and surgical and anesthetic management parameters of the patients studied.

Characteristic		Overall N = 562 (100%)	RA Success N = 514 (91.5%)	RA Failure N = 48 (8.5%)	*p*-Value ^a b^
Sex n (%)	Male	350 (100%)	327 (93.4%)	23 (6.57%)	0.04
	Female	212 (100%)	187 (88.2%)	25 (11.8%)	
Age (SD)		41.3 (16.2)	41.5 (16.1)	38.9 (16.4)	0.28
ASA Score n (%)	ASA I	404 (100%)	376 (93%)	28 (6.93%)	
	ASA II	150 (100%)	131 (87.3%)	19 (12.7%)	0.03
	ASA III	8 (100%)	7 (87.5%)	1 (12.5%)	
	ASA IV	0 (0%)	0 (0%)	0 (0%)	
Substance abuse n (%)		6 (100%)	3 (50%)	3 (50%)	<0.001
Depressive syndrome n (%)		23 (100%)	20 (86.9%)	3 (13.1%)	0.44
Chronic analgesic treatment n (%)		19 (100%)	17 (89.5%)	2 (10.5%)	0.67
Diagnosis before surgery, n (%)	Whitlow	370 (100%)	341 (92.2%)	29 (7.8%)	
	Door finger	89 (100%)	77 (86.5%)	12 (13.5%)	0.20
	Wound	103 (100%)	96 (93.2%)	7 (6.8%)	
Surgical area, n (%)	Ambulatory surgical unit	321 (100%)	302 (94.1%)	19 (5.9%)	0.01
	Conventional surgical unit	241 (100%)	212 (88%)	29 (12%)	
Anesthetist, n (%)	Resident	418 (100%)	385 (92.1%)	33 (7.9%)	
	Fellow	44 (100%)	37 (84.1%)	7 (145.9%)	0.21
	Assistant Professor	100 (100%)	92 (92%)	8 (8%)	
Average waiting time before RA, min (SD)		55.2 (60.8)	56.6 (61)	40.4 (57.3)	0.09
Block type, n (%)	Axillary block	275 (100%)	252 (91.7%)	23 (8.3%)	
	Truncular block	99 (100%)	89 (89.9%)	10 (10.2%)	0.61
	Digital block	20 (100%)	20 100%)	0 (0%)	
	AxB + truncular blocks	168 (100%)	153 (91.1%)	15 (8.9%)	
Local anesthetic n (%)	Lidocaine	297 (100%)	274 (92.3%)	23 (7.7%)	
	Ropivacaine	87 (100%)	77 (88.5%)	10 (11.5%)	0.51
	Both	178 (100%)	163 (91.7%)	15 (8.3%)	
Average time between RA and surgery, min (SD)		41.3 (35.5)	41.1 (34.2)	43.6 (47.8)	0.34
Period of the day	Night	45 (100%)	39 (86.7%)	6 (13.3%)	0.09
	Morning	357 (100%)	333 (93.3%)	24 (6.7%)	
	Afternoon	160 (100%)	142 (88.8%)	18 (11.2%)	

Abbreviations: SD: standard deviation; ASA: American Society of Anesthesiologists; min: minutes; AxB: axillary block; RA: regional anesthesia; mg: milligram. **a**: *p*-value compared to success group versus failure group. **b**: The values are expressed as an average (standard deviation) or actual (%). The *p*-value results from an exact Fisher test for qualitative variables and a Mann–Whitney test for quantitative variables whose distribution was not normal.

**Table 2 jcm-09-02453-t002:** Implication of regional anesthesia (RA) failure on the patient’s journey: analysis of surgical and anesthetic data during and after surgery.

Variable	RA Success N = 514 (91.5%)	RA Failure N = 48 (8.5%)	*p*-Value ^a b^
Average surgery time, min (SD)	26.6 (18.7)	35.5 (28.9)	0.02
Intraoperative analgesics (acetaminophen or nefopam), n (%)	18 (3.5%)	7 (14.6%)	<0.01
PONV, n (%)	3 (0.58%)	1 (2.08%)	0.30
Analgesics after surgery, n (%)	114 (22.2%)	11 (22.9%)	0.86
Average duration in recovery room, min (SD)	32.5 (19.3)	45.1 (26,4)	<0.01
Hospitalization, n (%)	22 (4.28%)	8 (16.7%)	<0.01

Abbreviations: min: minutes; SD: standard deviation; **a**: *p*-value comparing success group versus failure group; **b**: the values are expressed as median (standard deviation) or actual (%). The *p*-value results from an exact Fisher test for qualitative variables and a Mann–Whitney test for quantitative variables whose distribution was not normal.

**Table 3 jcm-09-02453-t003:** Hospitalizations according to the time of day when surgery was performed.

Period of Surgery	Hospitalizations		*p*
	No (n = 531)	Yes (n = 30)	
Morning (8 a.m. to 12 p.m.) n (%)	355 (67%)	1 (3%)	
Afternoon (12 p.m. to 6 p.m.) n (%)	155 (29%)	5 (17%)	*p* < 0.0001
Night (6 p.m. to 8 a.m.) n (%)	21 (4%)	24 (80%)	

The *p*-value results from an exact Fisher test.

**Table 4 jcm-09-02453-t004:** Multivariate analysis: significant risk factors for RA failure.

Variable	OR	IC 95%	*p*-Value
Sex: female	2.35	1.24–4.45	0.01
Conventional surgery unit	2.33	1.25–4.34	0.01
Substance abuse	7.27	1.18–44.88	0.03
ASA score	2.02	1.01–3.94	0.04

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
