# Peer review of "Impact of Regional Block Failure in Ambulatory Hand Surgery on Patient Management: A Cohort Study"

_jcm, 2020, doi:10.3390/jcm9082453_

Round 1

Reviewer 1 Report

The study would have been much improved if it had been performed prospectively. Retrospective studies suffer from potential bias, and lack of robust reporting. It is surprising that the level of training and experience of the staff did not play a role in the number of block failures. The majority of failures occurred in the Resident block pool. Though not statistically significant it would have been interesting data to have seen the ultra sound images so that poor technique could be clearly demonstrated. Going forwards a prospective audit of outcomes would provide much stronger data and demonstrate if lack of training and/or experience was a significant factor. This is all interesting information but to gain improvement in outcomes prospective audits should be in place.

Author Response

The study would have been much improved if it had been performed prospectively. Retrospective studies suffer from potential bias, and lack of robust reporting. It is surprising that the level of training and experience of the staff did not play a role in the number of block failures. The majority of failures occurred in the Resident block pool. Though not statistically significant it would have been interesting data to have seen the ultra sound images so that poor technique could be clearly demonstrated. Going forwards a prospective audit of outcomes would provide much stronger data and demonstrate if lack of training and/or experience was a significant factor. This is all interesting information but to gain improvement in outcomes prospective audits should be in place.

First of all, thank you very much for your interest in our study and the time you took to suggest ways to improve it. As mentioned in the limitation section, “this study is a retrospective, single center, observational study with possible bias. We included, however, an important number of patients. The surgical procedures were standardized and performed routinely, making the cohort of patients quite homogeneous. Furthermore, the collection of perioperative data was standardized and exhaustive”. All this makes it possible to reduce the number of biases inherent in retrospective studies.

Regarding the fact that there are no more failures according to the experience of the anesthesiologist, we explain this result by the fact that Saint Antoine hospital is an expert center in locoregional. Residents benefit from expert and close training. They are empowered very quickly, which allows a very quick learning curve. Upstream, there is also training on a mannequin which allows to quickly acquire the right gestures. Also, the majority of studies have shown that it only takes 20-30 blocks before being autonomous for performing this type of anesthesia [1,2]. Thanks to the predominantly regional anesthesia performed in Saint Antoine, this number of blocks is very quickly reached from the first week for residents. It would indeed be interesting to specify it in the discussion.

Although the majority of clinical studies focusing on risk factors are retrospective, we agree with the reviewer and consider that it would be really interesting to conduct a prospective study to confirm these results.

Reviewer 2 Report

Study design: The study was retrospective. How were the data collected? Electronic charts?

Study outcomes: When was sedation considered as a “real sedation” (due to anxiety) and when a block failure?

Statistical methods: How did you determine beforehand the number of patients or the time for period for data collecting needed for your study?

Table1 could be more informative. I would but with all variables the success rate and failure rate. Now you , for instance have the percentage of both sexes in the success and failure column. On the other hand, you have success and failure rates of the depressive patients (that is more informative). Please, correct.

What is the “waiting time” before RA?¨

Truncular block: It is not clear from the text (at least not for me). Was this performed with us? From which level?

Digital block: Was this also performed by an anaesthesiologist? Not with us? I think that you should exclude the digital block patients from your study. The digital blocks are mostly performed by the surgeon and are different from the blocks that are performed more proximally

AxB + truncular block: Was this a primary plan to make the approach from different parts of the brachial plexus? Or truncular block after a failure of axillary block? I think that inserting needle to a at least partially anaesthesied nerve carries a potential of neural damage unless the performer is very experienced.

Average doses of local anaesthetics are irrelevant if some of the patients received one local anaesthetic as a sole anaesthetic and some patients as a mixture of two

Period of the day: Please, define more precisely

3.2 I guess, it should be “block failure was associated…

I do not understand the statement that the patients with failure benefited more from analgesics than the patients with successful blocks. Was it not a failure according to your methods if the patient received analgesics? Please, explain.

Please, do not use the phrase “.., but this was  not significant…”. Either statistically or clinically significant.

Were the criteria for PACU stay and hospitalization standardized and similar during the whole study period? I guess that a patient operated 1a.m. is more often willing to stay in the hospital until the following day than a patient operated at 9 a.m.

How did you define “substance abuse”?

The discussion could be condensed and written more comprehensevily. You should concentrate on the relevant findings of your study and their relation to existing literature. For instance, the neuronal signaling chapter is irrelevant here.

Author Response

Study design: The study was retrospective. How were the data collected? Electronic charts?

First of all, we thank the reviewer for the time he took in reading our manuscript. The data were collected using electronic health records (anesthesia consultation, per and post-operative data). These electronic records are also kept in paper format in the hospital archives and are easily accessible.

Study outcomes: When was sedation considered as a “real sedation” (due to anxiety) and when a block failure?

The definition of the French Anesthesiology and Intensive Care Medicine Society (SFAR) regarding block failure does not retain a threshold dose for the use of hypnotic drugs. Under these conditions, as soon as a sedation was administered, the block was considered to be a failure. Our center has expertise in ambulatory care as the first surgical center for the upper limb of the Assistance Publique Hôpitaux de Paris. Our anesthetists and nurse anesthetists are trained in the management of intra-operative anxiety by avoiding the use of hypnotic or anxiolytic drugs.

Statistical methods: How did you determine beforehand the number of patients or the time for period for data collecting needed for your study?

This study aimed at identifying factors associated with block failure and a preliminary sample size calculation assessed that about 50 events would be necessary to identify the main risk factors with sufficient power. After two years of recruitment, our sample size (48 patients with bloc failure and 514 without) allowed us to identify an OR of about 2.5 for a 20 to 50% proportion of exposed controls with an 80% power, which we considered enough to end recruitment. For more clarity, we add this sentence in the Methods section.

Table1 could be more informative. I would but with all variables the success rate and failure rate. Now you , for instance have the percentage of both sexes in the success and failure column. On the other hand, you have success and failure rates of the depressive patients (that is more informative). Please, correct.

We are sorry, but we are not sure to understand the question from the reviewer. We tried to compare all the patient’s characteristics that we obtained from the health records and that may impact the regional anesthesia quality.

What is the “waiting time” before RA?¨

“Waiting time before RA” is the time from the patient’s arrival in the surgical unit, and the realization of the anesthesia. This “waiting time” depends on the operating room program. For more clarity, this definition was added to the Methods section.

Truncular block: It is not clear from the text (at least not for me). Was this performed with us? From which level?

The truncular block is performed by the anesthesiologist, either in a single block or in addition to an axillary block. This involves directly blocking the median, radial or ulnar nerve distally (mainly at the forearm) using ultrasound. The choice of anesthesia technique is defined by the anesthesiologist according to the gesture to be performed and the extent of the surgery to one or more nervous areas.

Digital block: Was this also performed by an anaesthesiologist? Not with us? I think that you should exclude the digital block patients from your study. The digital blocks are mostly performed by the surgeon and are different from the blocks that are performed more proximally

All loco-regional or local blocks are always performed by the anesthesiologist and not by the surgeon at Saint Antoine hospital, including digital blocks. Indeed, when the foreseeable surgical gesture is not very broad and concerns the distal level of the fingers, we offer this type of anesthesia. For this reason, we decided to include these patients concerned, even if it is a minority (20 patients among 562 or 3.56%).

AxB + truncular block: Was this a primary plan to make the approach from different parts of the brachial plexus? Or truncular block after a failure of axillary block? I think that inserting needle to a at least partially anaesthesied nerve carries a potential of neural damage unless the performer is very experienced.

When the surgical procedure is at risk of lasting (complex surgery), or when the foreseeable post-operative pain is great, the axillary block is completed by one or more truncular blocks. In this case we use Ropivacine ®, as a long-acting anesthetic to provide qualitative analgesia without motor blockage of the entire upper limb. Indeed, the axillary block is quite always done with Xylocaine ®. A distinction must be made between complementary blocks performed preoperatively and blocks performed during surgery or postoperatively, a posteriori. In the latter case, the block is performed because of the pain complained of by the patient, and thus corresponds to a potential failure of the first block. To avoid any risk of neural damage, all our blocks are done using ultrasound guidance.

Average doses of local anaesthetics are irrelevant if some of the patients received one local anaesthetic as a sole anaesthetic and some patients as a mixture of two

We thank the reviewer for this comment. To gain clarity, we deleted the row in table 1.

Period of the day: Please, define more precisely

To be more precise, the night corresponds to the period from 6 p.m. to 12 a.m., the morning to the period from 8 a.m. to 12 p.m. and finally the afternoon to the period from 12 p.m. to 6 p.m. We precised these definitions in Table 4.

I guess, it should be “block failure was associated…

We thank the reviewer for this comment. We changed the sentence as recommended.

I do not understand the statement that the patients with failure benefited more from analgesics than the patients with successful blocks. Was it not a failure according to your methods if the patient received analgesics? Please, explain.

Patients who had failed loco-regional anesthesia required additional analgesia to manage their pain intraoperatively compared to the “success” group. Indeed, the failure of the loco-regional anesthesia first manifests itself by a painful complaint from the patient. This is a likely consequence of the failure without a causal link being asserted. The administration of analgesics is at the discretion of the anesthesiologist managing the patient and is not a criterion in the definition of RA failure according to the French Anesthesiology and Intensive Care Medicine Society (SFAR). Conversely, only the need for additional block, sedation, or GA are the criterion of RA failure.

Please, do not use the phrase “.., but this was not significant…”. Either statistically or clinically significant.

We thank the reviewer for this comment. We changed the sentence as recommended.

Were the criteria for PACU stay and hospitalization standardized and similar during the whole study period? I guess that a patient operated 1a.m. is more often willing to stay in the hospital until the following day than a patient operated at 9 a.m.

All surgeries were performed on an outpatient basis, regardless of the part of the day. We authorize outings in this context at night, the only condition being the possibility for the patient to have a companion. It is indeed a potential bias but by chance it concerns a minority of patients (45 patients or 8%, operated before 12 a.m.).

How did you define “substance abuse”?

The history of drug addiction was defined through anesthesia consultation. The patient was considered to have a history or an active substance abuse in the case of declared drug use, or of substance substitution.

The discussion could be condensed and written more comprehensevily. You should concentrate on the relevant findings of your study and their relation to existing literature. For instance, the neuronal signaling chapter is irrelevant here.

We thank the reviewer for this comment. We modified the discussion as recommended, hoping that this will meet the reviewer's expectations. 

Reviewer 3 Report

This is a retrospective single-center observational cohort study to investigate the causes and consequences of regional anesthesia (RA failure) in an ambulatory care center and associated hospital in Paris, metropolitan France.

While the study has moderate attrition, is well documented and methodologically and statistically relatively sound, I am underwhelmed by the lack of novelty and clinical impact of the findings and conceptual framework.

As a regional anesthesiologist, many questions remain after reading this manuscript. Why did the patients in the successful RA group require so often analgesia in the PACU (28%)? The most important factor for RA failure may be operator competence?? The manuscript states that this could be excluded, but was this solely based on the level of training of the person holding the needle, or did they investigate if different supervising attending anesthesiologists had different failure rates? Also, the detection bias is a concern as some clinicians will just wiggle through a failed RA, while other immediately convert to a clean LMA/GA with Propofol TIVA? It is unclear to me why were there so many axillary blocks (about 50%) if ultrasound was available? Would a supraclavicular block not be faster in onset, more reliable lasting longer… Is “block failure” actually really a failure, because GA is extremely safe and with Propofol extremely well-tolerated, hence the postoperative pain control is more important as a benefit of RA? Why not just perform the block, take the patient to the OR immediately before the block has time to set and administer GA with Propofol and an LMA, reducing PONV and leading to very short PACU stays?

To improve the manuscript, I would recommend explaining the mechanistic concept the author had a priori about why the various predictors would lead to higher failure rates. A better explanation as to the odd clinical choices (50% axillary blocks) and the unclear allocation to night versus daytime surgery would help. As a patient, I may not want to leave the hospital at midnight right after my surgery was completed, but that is commonplace in Paris?

In summary, while the study is formally well executed, it is unclear to me how this study will inform my clinical practice and improve patient care or safety.

Author Response

Restitution of questions

First of all, we thank the reviewer for the time he took in reading our manuscript allowing us to improve its quality.

Why did the patients in the successful RA group require so often analgesia in the PACU (28%)?

The use of analgesics in the PACU was 22.2% in the success group and 22.9% in the failure group (p O, 86 Table 3.). We use a standard protocol of analgesics at Saint Antoine hospital in case of pain. The resorption kinetic of local anesthetics, in particular short-acting molecules (Xylocaine ®) leads to a decrease of the block efficiency during the post-operative period. It is therefore necessary to take over with conventional analgesic drugs at that time.

The most important factor for RA failure may be operator competence?? The manuscript states that this could be excluded, but was this solely based on the level of training of the person holding the needle, or did they investigate if different supervising attending anesthesiologists had different failure rates?

At Saint Antoine Hospital, a team of 6 senior anesthesiologists perform anesthesia for ambulatory surgery. These doctors also train the residents in a close and intense way. Our data do not allow to highlight any significant difference in terms of failure associated with the experience of the anesthesiologist. Saint Antoine hospital, as the first surgical center for the upper limb of the Assistance Publique Hôpitaux de Paris allows residents to benefit from expert and close training. They are empowered very quickly, which allows a very quick learning curve. Upstream, they are also trained on a mannequin which allows to quickly acquire the right gestures. Majority of studies have shown that rates of failure in loco-regional is largely reduced after 20 to 30 blocks [1,2]. Thanks to the predominantly regional anesthesia performed in Saint Antoine, this number of blocks is very quickly reached from the first week for residents.

Also, the detection bias is a concern as some clinicians will just wiggle through a failed RA, while other immediately convert to a clean LMA/GA with Propofol TIVA?

We completely agree with this remark, unfortunately it is an inherent bias in the study given the definition of failure given by the French Anesthesiology and Intensive Care Medicine Society (SFAR). But considering that our practice is homogenous, as the team of senior anesthesiologists remains the same since many years and as the first surgical center for the upper limb of the Assistance Publique Hôpitaux de Paris, we consider that this detection bias exists but should be considered as marginal. We add a sentence to gain clarity in the discussion part of our manuscript.

It is unclear to me why were there so many axillary blocks (about 50%) if ultrasound was available? Would a supraclavicular block not be faster in onset, more reliable lasting longer… Is “block failure” actually really a failure, because GA is extremely safe and with Propofol extremely well-tolerated, hence the postoperative pain control is more important as a benefit of RA?

In the case of ambulatory surgery in France, the supraclavicular block is not recommended by the French Anesthesiology and Intensive Care Medicine Society (SFAR) due to the risk of pneumothorax. distal surgery of the upper limb, the axillary block is recommended in first intention. All the blocks are done under ultrasonography. Block failure, as the SFAR defined it, is a real failure regardless of the risk of GA, that we agree is really low. But to obtain the clearest results, we considered a strict definition of block failure.

Why not just perform the block, take the patient to the OR immediately before the block has time to set and administer GA with Propofol and an LMA, reducing PONV and leading to very short PACU stays?

The objective of our team is to reduce the use of GA, considering that several studies document the benefits of RA when compared to general anesthesia [3,4] in terms of post-operative analgesia and postoperative opioid consumption. Considering the patient flow, our goal is to make the block as close as possible to the patient’s transfer to the OR. We start anesthesia in a dedicated room when the surgeon is about to complete his previous intervention.

To improve the manuscript, I would recommend explaining the mechanistic concept the author had a priori about why the various predictors would lead to higher failure rates. A better explanation as to the odd clinical choices (50% axillary blocks) and the unclear allocation to night versus daytime surgery would help. As a patient, I may not want to leave the hospital at midnight right after my surgery was completed, but that is commonplace in Paris?

We thank the reviewer for this constructive suggestion. All surgeries were performed on an outpatient basis, regardless of when the patient left. We authorize outings in this context at night, the only condition being the possibility for the patient to have a companion. It is indeed a potential bias but by chance it concerns a minority of patients (45 patients or 8%, operated before 12 a.m.). We completed the discussion with this element.

In summary, while the study is formally well executed, it is unclear to me how this study will inform my clinical practice and improve patient care or safety.

The major highlight of the current manuscript, based on a monocentric cohort study, is that block failure significantly increased the incidence of unplanned hospitalizations and the Post-Anesthesia Care Unit length of stay. To our knowledge, it is also the first study to precise the risk factors of block failure since the rise of ambulatory care and ultrasonography. Knowing and seeking these, patient care can be improved.

Round 2

Reviewer 2 Report

Table1 is very busy and hard to understand. Sorry, for the typing error in my previous comments. I would put in parentheses the row percentage with all variables with the success rate and failure rate. Now you, for instance have the percentage of both sexes in the success and failure column. On the other hand, you have success and failure rates of the depressive patients in a row (that is more informative). Please, correct. An example as it should be in my opinion:

                    Total                       Success                        Failure

Sex n (%) Male 350 (62.3%)       327 (93.4%)                  23 (6.6%)

                Female 212 (37.7%)   187 (88.2%)                  25 (11.8%)

Then you see at one glance the success and failure rates with both sexes.

Also in Table 1: Please, remove "%" from "Average waiting time min" and RA success.

Table 1. LRA not explained in the abbreviations.

Table 2. Why do you present the positive and negtive risk factors in the same table? The Title of the Table is significant risk factors for failure. Then you have two positive and twio negative factors in the tTable. Please, insert only "positive" risk factors like female sex or change the title of the Table.

You did not answer earlier if you used ultrasound with your digital blocks (I guess not). If you did not, your statement about block failure rate with ultrasound in the "Discussion" is not exact. Therefore, I still thinkthat this patient group is so different frrom the other patients that it should be omitted and the statitics recalculated accordingly.

Author Response

Table1 is very busy and hard to understand. Sorry, for the typing error in my previous comments. I would put in parentheses the row percentage with all variables with the success rate and failure rate. Now you, for instance have the percentage of both sexes in the success and failure column. On the other hand, you have success and failure rates of the depressive patients in a row (that is more informative). Please, correct. An example as it should be in my opinion:

                    Total                       Success                        Failure

Sex n (%) Male 350 (62.3%)       327 (93.4%)                  23 (6.6%)

                Female 212 (37.7%)   187 (88.2%)                  25 (11.8%)

Then you see at one glance the success and failure rates with both sexes.

First of all, thank you very much for your precise review of our manuscript. As recommended, we have corrected all the percentages of Table 1 considering the row and not the group that is concerned by the data. We hope that will improve the clarity of this table.  

Also in Table 1: Please, remove "%" from "Average waiting time min" and RA success. We are sorry but we do not see this mistake in Table 1. Unless we are wrong, each “average waiting time xxx, min” is following by SD (standard deviation) values. Table 1. LRA not explained in the abbreviations. This is now corrected.  Table 2. Why do you present the positive and negtive risk factors in the same table? The Title of the Table is significant risk factors for failure. Then you have two positive and twio negative factors in the tTable. Please, insert only "positive" risk factors like female sex or change the title of the Table. 

We thank the reviewer for his remark. We have now corrected the table 2 and the descriptive part of the results section in order to obtain only “risk factors” that are associated with an increased risk to belong to the “failure” group. 

You did not answer earlier if you used ultrasound with your digital blocks (I guess not). If you did not, your statement about block failure rate with ultrasound in the "Discussion" is not exact. Therefore, I still thinkthat this patient group is so different frrom the other patients that it should be omitted and the statitics recalculated accordingly. 

Our digital blocks are not realized using ultrasound guidance but considering that i/ there is only 3.5% of patients that benefit of that type of block, ii/ this block is recommended in some types of hand surgery and iii/ no block failure was observed in this group, we consider that it is important to preserve this kind of patients in our analyses. But we agree with the reviewer comment and that explains our sentence in the discussion: “The incidence of failure may depend on the type of block performed and the technique of performance”. We hope that this explanation will support you in the way we have chosen to present the results.

Reviewer 3 Report

I again find that the study is methodologically well-conducted, but I take issue with the fundamental premise underlying this investigation and question how it would inform an international perioperative readership.

This study asks the wrong questions.

The presented clinical practice is alien to what is common-place in the US and makes little sense from my perspective. The contrast of General Anesthesia versus Regional Anesthesia as a strict alternative in otherwise reasonably healthy patients is outdated. Instead of focusing on intra-op management, [which often is inconsequential], we should consider the entire perioperative continuum, in particular long-term outcomes, cost, and patient experience after discharge. So if we employ regional anesthesia, it should improve pain control after hospital discharge or improve rehabilitation. But the presented practice is not using long-acting local anesthetics?? Why not?

Furthermore, I disagree with the argument that supraclavicular blocks are too dangerous if done under ultrasound guidance. Alternatively, an infraclavicular block is a suitable alternative?

But even if we accept the premise of the authors, I cannot follow their protocols. In the age of ultrasound and with on average 40min delay between block and surgery, is there not enough time to test if the block is working adequately? If not, then is there not enough time for a rescue block of that particular nerve that is not blocked well enough?

However, in my experience and according to the literature, even a perfect regional anesthetic, especially an axillary block, may fail due to tourniquet pain. Did I miss the discussion of this in the manuscript?

Even if we follow the authors that it would be interesting to identify causes for block failure, this study does not answer the questions that matter most. An axillary approach to the brachial plexus for hand and lower arm surgery should block four nerves. An investigation into causes for block failure should seek to identify which of these nerves is often not sufficiently blocked (radial nerve?!) or why the block is insufficient for surgery (tourniquet pain?!) to come up with actionable recommendations of what we can do better. 

Avoiding surgery in the afternoon and at night seems not a viable option, does it? The difference between ambulatory versus outside the ambulatory surgery center may be statistically significant, but is likely clinically irrelevant: what is the number needed to treat to make a difference as in one patient not having to undergo GA and does it really matter to this patient if we just administer TIVA and LMA?

Author Response

I again find that the study is methodologically well-conducted, but I take issue with the fundamental premise underlying this investigation and question how it would inform an international perioperative readership.

This study asks the wrong questions.

First of all, thank you very much for his remarks on our manuscript and our methodology. At a time when locoregional anaesthesia (LRA) is a practice that has become widespread and even recommended [1], it seemed to us legitimate to look into the factors associated with its failure, considering that few recent studies have focused on it, although the method of LRA has evolved considerably since the development of ultrasound guidance. In our study, the failure rate is not null with 8.5% of LRA failure. Considering the importance of these failure on the patient’s flow in ambulatory setting, we consider that the questions asked in this study are of most importance for the international perioperative readership of Journal of Clinical Medicine, and that the identification of potentially related risk factors would be of great help for patient’s management.

The presented clinical practice is alien to what is common-place in the US and makes little sense from my perspective. The contrast of General Anesthesia versus Regional Anesthesia as a strict alternative in otherwise reasonably healthy patients is outdated. 

We fully agree with the reviewer that the anesthetic practices could be different from one country to another, but the interest of LRA compared to general anaesthesia (GA) seems quite clear, especially in orthopaedic surgery. As shown in various recent clinical studies and meta-analyses from Europe, Canada or China, and although there is some debate about its interest in preventing post-operative delirium, LRA appears to have hudge advantages in terms of analgesia, post-operative nausea and fast post-operative recovery compared to general anaesthesia (GA), whatever the surgery concerned [2–5].  

Instead of focusing on intra-op management, [which often is inconsequential], we should consider the entire perioperative continuum, in particular long-term outcomes, cost, and patient experience after discharge. 

We are precisely interested here in the whole perioperative continuum, taking into account the pre, per, and post-operative data for each patient. By their impact on the patient’s flow, including the length of PACU stay or unscheduled hospitalizations, locoregional anesthesia failures has important consequences on the financial and organizational cost for ambulatory surgery. But as it was shown in different studies [6,7], we confirm that anesthetic management necessarily influences the postoperative outcomes. By this point of view, we completely agree with the reviewer.

So if we employ regional anesthesia, it should improve pain control after hospital discharge or improve rehabilitation. But the presented practice is not using long-acting local anesthetics?? Why not?

This is what we try to demonstrate here considering failed LRA. Patients who have had LRA failure have a greater consumption of analgesics and they also stay longer in PACU. This was indicated in the results part: “These same patients also benefited from more frequent administration of intraoperative analgesics (18 patients (3.5%) in the success group vs 7 patients (14.6%) in the failure group, (p=0.01))” and “The duration of management in PACU was 39% longer in the failure group in comparison to the success group (45 min vs 32 min respectively) (p=0.0001)”.

However, we consider that the use of long-acting anesthetics (which is performed in almost 50% of our patients) is only necessary for surgeries which induce significant pain in the long term, which is not the case for example for whitlow. The essential limit of blocks with long duration anesthetics is the consequences in terms of motor block, that could induce functional limitations in patients. 

Furthermore, I disagree with the argument that supraclavicular blocks are too dangerous if done under ultrasound guidance. Alternatively, an infraclavicular block is a suitable alternative? 

We completely agree with the reviewer but in France these two blocks are for now contraindicated for ambulatory patients by the High Health Authority (HAS), because of the significant risks of pneumothorax, even if this recommendation is based on studies using neuro-stimulation.  

But even if we accept the premise of the authors, I cannot follow their protocols. In the age of ultrasound and with on average 40min delay between block and surgery, is there not enough time to test if the block is working adequately? If not, then is there not enough time for a rescue block of that particular nerve that is not blocked well enough? 

The blocks are obviously tested before transfer to the operating room, following international guidelines, by testing different sensitive territories using the light touch assessment [8]. However, it sometimes happens, even if the light touch assessment is positive before the surgery, that the patient feels pain when the incision is made. That could be partly explained by the emotional component of pain that is not covered by LRA, as we discussed that in our manuscript. In case of pre-operative LRA failure, an additional block or a change of strategy is then considered. But, for the majority of patients, the observation of failure was made before entering the operating room. 

However, in my experience and according to the literature, even a perfect regional anesthetic, especially an axillary block, may fail due to tourniquet pain. Did I miss the discussion of this in the manuscript? 

We agree ith this reviewer's remark, the axillary block cannot perfectly manage the pain induced by the tourniquet. That’s why we always coupled it with a subcutaneous injection to anesthetize the medial cutaneous nerve of the arm. We added a sentence in the method part to clarify this block.

Even if we follow the authors that it would be interesting to identify causes for block failure, this study does not answer the questions that matter most. An axillary approach to the brachial plexus for hand and lower arm surgery should block four nerves. An investigation into causes for block failure should seek to identify which of these nerves is often not sufficiently blocked (radial nerve?!) or why the block is insufficient for surgery (tourniquet pain?!) to come up with actionable recommendations of what we can do better.

We thank the reviewer for this remark. Our question was not to focus on the regional blocks that have been performed for several decades under the supervision of experienced anesthesiologists, but rather, in this context, to analyze the perioperative factors that can participate in LRA failures. These are obviously important questions and may be the objectives of further prospective study.

Avoiding surgery in the afternoon and at night seems not a viable option, does it?

This type of surgery is rarely an immediate emergency and can therefore be postponed. In our opinion, the message is precisely to avoid performing this surgery outside working hours in order to be able to prevent LRA failure and to avoid an unscheduled hospitalization. Even if the ambulatory surgery program is extremely dense, a specific flow should be implemented in all ambulatory structures to limit these situations.

The difference between ambulatory versus outside the ambulatory surgery center may be statistically significant, but is likely clinically irrelevant: what is the number needed to treat to make a difference as in one patient not having to undergo GA and does it really matter to this patient if we just administer TIVA and LMA?

Assuming a failure proportion of 12% in patients from conventional surgical unit and a multivariable OR of 2.33 (95% CI: 1.25, 4.34) when compared with ambulatory surgical unit, the number needed to treat (i.e. to switch from conventional to ambulatory surgical unit) to avoid one additional failure would be 2.9 (95% CI: 1.4, 7.5). However, this NNT assumes a causal relationship between surgical area and risk of failure, which this observational study is unable to demonstrate. For more clarity, we add a sentence resuming this information in the discussion part.

And considering all of the literature, cited above, comparing loco-regional anesthesia and general anesthesia, and following the French recommendations, we think that the benefit/risk ratio is largely in favor of loco-regional anesthesia in the surgical cases presented in our study.

We hope that these answers will support you in the way we have chosen to present the results.

  1. Neal, J.M.; Gerancher, J.C.; Hebl, J.R.; Ilfeld, B.M.; McCartney, C.J.L.; Franco, C.D.; Hogan, Q.H. Upper extremity regional anesthesia. Essentials of our current understanding, 2008. Reg. Anesth. Pain Med. 2009, 34, 134–170.
  2. Ilfeld, B.M.; Morey, T.E.; Wang, R.D.; Enneking, F.K. Continuous popliteal sciatic nerve block for postoperative pain control at home: A randomized, double-blinded, placeb-controlled study. Anesthesiology 2002, 97, 959–965, doi:10.1097/00000542-200210000-00031.
  3. Chery, J.; Semaan, E.; Darji, S.; Briggs, W.T.; Yarmush, J.; D’Ayala, M. Impact of regional versus general anesthesia on the clinical outcomes of patients undergoing major lower extremity amputation. In Proceedings of the Annals of Vascular Surgery; Elsevier Inc., 2014; Vol. 28, pp. 1149–1156.
  4. Ahn, E.J.; Kim, H.J.; Kim, K.W.; Choi, H.R.; Kang, H.; Bang, S.R. Comparison of general anaesthesia and regional anaesthesia in terms of mortality and complications in elderly patients with hip fracture: A nationwide population-based study. BMJ Open 2019, 9, doi:10.1136/bmjopen-2019-029245.
  5. Hopkins, P.M. Does regional anaesthesia improve outcome? Br. J. Anaesth. 2015, 115, 26–33, doi:10.1093/bja/aev377.
  6. Monk, T.G.; Saini, V.; Weldon, B.C.; Sigl, J.C. Anesthetic management and one-year mortality after noncardiac surgery. Anesth. Analg. 2005, 100, 4–10, doi:10.1213/01.ANE.0000147519.82841.5E.
  7. Sessler, D.I. Long-term consequences of anesthetic management. Anesthesiology 2009, 111, 1–4.
  8. Ode, K.; Selvaraj, S.; Smith, A.F. Monitoring regional blockade. Anaesthesia 2017, 72, 70–75, doi:10.1111/anae.13742.
